# Spatial and Temporal Gene Function Studies in Rodents: Towards Gene-Based Therapies for Autism Spectrum Disorder

**DOI:** 10.3390/genes13010028

**Published:** 2021-12-23

**Authors:** Iris W. Riemersma, Robbert Havekes, Martien J. H. Kas

**Affiliations:** Groningen Institute for Evolutionary Life Sciences, Neurobiology, University of Groningen, Nijenborgh 7, 9747 AG Groningen, The Netherlands; i.w.riemersma@rug.nl (I.W.R.); r.havekes@rug.nl (R.H.)

**Keywords:** autism spectrum disorder, animal models, synaptopathology, neurodevelopment, sensory processing

## Abstract

Autism spectrum disorder (ASD) is a complex neurodevelopmental condition that is characterized by differences in social interaction, repetitive behaviors, restricted interests, and sensory differences beginning early in life. Especially sensory symptoms are highly correlated with the severity of other behavioral differences. ASD is a highly heterogeneous condition on multiple levels, including clinical presentation, genetics, and developmental trajectories. Over a thousand genes have been implicated in ASD. This has facilitated the generation of more than two hundred genetic mouse models that are contributing to understanding the biological underpinnings of ASD. Since the first symptoms already arise during early life, it is especially important to identify both spatial and temporal gene functions in relation to the ASD phenotype. To further decompose the heterogeneity, ASD-related genes can be divided into different subgroups based on common functions, such as genes involved in synaptic function. Furthermore, finding common biological processes that are modulated by this subgroup of genes is essential for possible patient stratification and the development of personalized early treatments. Here, we review the current knowledge on behavioral rodent models of synaptic dysfunction by focusing on behavioral phenotypes, spatial and temporal gene function, and molecular targets that could lead to new targeted gene-based therapy.

## 1. Introduction

Autism spectrum disorder (ASD) is a complex lifelong neurodevelopmental condition that is characterized by difficulties with social interaction and communication, and by repetitive, restricted, and stereotyped behaviors. ASD is a highly heterogeneous condition on multiple levels including clinical presentation, genetics, and developmental trajectories [1]. The presentation and severity of these symptoms is highly diverse in autistic people and is often accompanied by comorbidities [2]. ASD is generally diagnosed during early life. First symptoms might arise at 6 months of age even before diagnosis generally takes place [3,4]. This highlights the developmental character of ASD. During this period in early life, brain development is heavily influenced by sensory experience [5]. Sensory symptoms especially can be found during early development as early as at 3 months of age [3]. This group of symptoms is characterized by hyper- or hyposensitivity to sensory stimuli, which can be of every modality such as increased or reduced sensitivity to light, sound, color, smell, taste, or touch [6]. Sensory symptoms are a common feature of ASD and can be found in about 90% of autistic people [7,8]. Since the inclusion of sensory symptoms in the DSM-V, more focus has gone towards the underlying neurobiology of those sensory symptoms. They are highly correlated with the diagnosis and severity of ASD and seem to be a predictor of social and repetitive symptom severity [3,9,10]. This underscores the importance of sensory symptoms and the need to further examine the underlying biological mechanisms of sensory symptoms.

Research into the underlying causes of ASD has revealed a high level of heritability [11,12,13,14,15]. Related to this, a large number of genetic studies have identified over a thousand genes linked to ASD, which have been uploaded to an online genetic database from the Simons Foundation Research Initiative (SFARI) [16]. The genetic architecture of ASD is very diverse and can derive from common or rare genetic variants, inherited or de novo variants, and different types of variants such as copy number variations (CNVs) or single nucleotide polymorphism (SNP) [17]. In addition to monogenetic causes, polygenic variations also contribute to ASD vulnerability. These genetic variations are involved with varying strengths of evidence and varying levels of penetrance reflecting the heterogeneity of the genetics in ASD and revealing the strong but diverse genetic component [16,18]. Despite the genetic heterogeneity, these genes cluster in groups with similar functions. Two major clusters are genes involved in gene regulation by influencing protein translation or transcription (*MECP2*, *CHD8*, *PTEN*, *FMR1*), and genes involved in synaptic function (*NRXN1*, *NRLGN*, *CNTNAP2*, *SHANK3*) [19,20]. Such gene clusters could possibly reveal convergence in biological functions and underlying neurodevelopmental processes [19]. Of the above-mentioned clusters, the synaptic proteins are of particular interest, as they are heavily involved in shaping synaptic connections by regulating synaptic formation, function, and maintenance. Establishing such neuronal connections is especially important during development when functioning brain circuits are formed. This group of synaptic proteins includes various cell adhesion molecules such as the presynaptic neurexin 1 (NRXN1) and postsynaptic neuroligin (NRLGN) that anchor in the synaptic membrane and bind each other to create stable trans-synaptic connections. In addition, other synaptic proteins belong to this group such as SHANK3 which functions as an anchor to membrane proteins and receptors in the synapse. In this way, these proteins are essential for normal synapse formation, maintenance, and functioning.

A major challenge is to understand how these gene clusters influence processes that lead to behavioral differences in ASD and when during development these changes occur. Another major question is if these genes also lead to a common phenotype during development through acting on shared biological mechanisms. Because of the heterogeneity on the levels of genetics, clinical presentation, and developmental trajectories, it is important to find such commonalities. No validated biomarkers and limited effective symptomatic or condition-modifying therapeutics exist to date [21]. Understanding these biological mechanisms is therefore important for potential biomarker discovery and the development of targeted interventions. Animal models can be utilized to study spatial and temporal gene function of ASD-related genes on multiple levels including at the level of behavior, brain circuits, and molecular function. Here, we discuss current findings on animal models of ASD-linked genes. We primarily focus on ASD-linked genes that contribute to synaptic dysfunction. First, we will describe the use of genetic animal models to study behavioral trajectories, including sensory symptoms, related to ASD. Secondly, spatial and temporal gene function will be discussed at the brain circuit and molecular level. Finally, we will address potential new targeted gene-based therapy based on findings in genetic animal models.

## 2. From Human ASD Genetics to Translational Behavioral Phenotype in Mouse Models

Despite the discovery of over a thousand genes implicated in ASD, a lot remains unknown about how these genes lead to an ASD phenotype. The high degree of heritability in ASD makes it suitable to study the effect of gene function using genetic knockout models. The advances in genetic discoveries in ASD have facilitated the generation of over 200 genetic ASD mouse models [16]. As mentioned previously, animal models are essential to give more insight into the relationship between gene function and behavioral phenotypes. Indeed, mouse models specifically have been proven essential for this because of the wide ability of genetic manipulation to study genes of interest in mice [22]. In addition, the shorter life span makes animal models an important tool to give insight into developmental trajectories. Given the strong genetic basis of ASD, one effective approach is to look at a cluster of ASD genes with similar functions to see if convergence in gene function also leads to convergence in biological processes underlying ASD phenotypes. Here, we will describe the current findings using the examples of three established genetic mouse modes of synapse dysfunction, *Nrnx1*, *Cntnap2*, and *Shank3*, which are all linked to ASD [16]. These three genes are evolutionary conserved and have essential roles in synaptic function across species thus making them suitable to study in mouse models [23,24,25,26,27]. We will first describe the current findings at the behavioral level in these models. Next, we will go into the underlying biology by discussing the expression patterns of these genes throughout specific brain regions and specifically during development. Subsequently, we will describe the molecular pathways in which these genes are involved.

### 2.1. Behavioral Phenotypes in Genetic Mouse Models of Synaptic Dysfunction in ASD

Core symptoms of ASD can be found in mice with genetic mutations in ASD-related genes [2]. One of the core symptoms in ASD are difficulties in social communication and interaction such as differences in social approach or verbal and nonverbal communication [6]. To assess behavioral phenotypes in animal models, many standardized tests exist [2,28]. The three-chamber task is for example ideally suited to assess sociability and social novelty. Furthermore, recordings of ultrasonic vocalization (USV) can give valuable information on social communication impairments from pup age onwards. The second main behavioral domain in ASD is the expression of restricted, repetitive, and stereotyped behaviors such as repetitive motor movements or behavioral inflexibility [6]. In rodent models, several behavioral tasks exist to study this domain for example self-grooming, marble burying, and the monitoring of stereotyped or repetitive locomotor activity in the open field [2]. Since the inclusion of sensory symptoms in the DSM-V, more attention is given to the assessment of sensory phenotypes, which present as hyper- or hyporeactivity to sensory input or unusual interest in certain sensory aspects in the environment [6]. To assess sensory phenotypes in rodent models, responses to sensory stimuli can be measured such as in the von Frey test for mechanical sensitivity and the hotplate test for heat responsiveness [28]. Differences in sensory processing can be examined by paired-pulse tasks such as pre-pulse inhibition [29]. In addition to these core features of ASD, autistic people show high rates of comorbidities such as attention deficit hyperactivity disorder, intellectual disability, epilepsy, anxiety, and depression [2]. Mouse models display phenotypes related to these comorbidities such as decreased attention, motor deficits, cognitive deficits, epileptic seizures, increased aggression, and anxiety-like behaviors [30,31,32,33,34,35,36,37,38,39,40,41,42]. Such comorbidity-related behaviors can be assessed by the open field, rotarod, or the elevated plus maze task [2,28], among others. We will discuss the phenotypes across the key domains of ASD; sensory, repetitive and social behaviors, and development in the *Nrnxn1*, *Cntnap2*, and *Shank3* mouse models of synaptic dysfunction (Table 1). Behavioral data in mouse models is variable, partly because of the use of different genetic isoforms; therefore, not all negative results are included that can be found in the current literature.

Mutations in the *NRXN1* gene are associated with ASD [43,44,45]. The *NRXN1* gene codes for an adhesion molecule on the presynaptic terminal that binds to neuroligins [44]. Two predominant isoforms exist, *NRXN1α* and *NRXN1β*. Mice lacking *Nrxn1* exhibit deficits in core symptoms related to ASD. Specifically, *Nrxn1α* mutants display deficits in the social domain including increased social preference, decreased social investigation, decreased social interaction, and increased aggression [35,46,47,48]. Furthermore, they show stereotypic and repetitive behaviors such as increased grooming, impaired nest-building, altered novelty responsiveness, and habituation [31,35,38]. In addition, several behaviors were reported that relate to ASD comorbidities including deficits in learning and cognition, locomotor activity, motor performance, award processing, and anxiety-like behaviors [31,35,36,38,46,47,48,49]. Although the social, stereotyped, and repetitive behaviors are commonly studied, sensory behaviors have received less attention in this model. *Nrxn1α* mice exhibit no deficits in olfactory processing. Since social behaviors heavily rely on olfactory cues, the observed differences in social behavior might therefore not be due to deficits in olfaction [47]. However, *Nrxn1* knockout mice show increased responses to auditory stimuli, seen by increases in startle responses to 80–100 dB auditory stimuli and increased startle response to 120 dB startle, indicating altered auditory sensitivity [31,36] In addition, while acoustic pre-pulse inhibition does not seem to be affected in heterozygous knockout mice or rat model [46], homozygous knockout mice display a decrease in pre-pulse inhibition [35,46,47]. Lastly, *Nrxn1* knockout rats display decreased eye-blink conditioning, indicating altered cerebral sensory learning [48]. There was no indication of differences in olfaction, while other sensory domains, such as tactile and visual sensitivity, still remain to be investigated. Together, these findings suggest that *Nrnx1* knockout models exhibit decreased cerebellar sensory learning and auditory sensitivity and processing deficits while also displaying deficits in the two core domains: social interactions and stereotyped and repetitive behaviors.

Contactin-associated protein 2 (CASPR2), the protein product of *CNTNAP2,* is a transmembrane adhesion molecule belonging to the neurexin superfamily of proteins that interacts with contactin-2 (CNTN2) to establish synaptic transmission [50]. Loss of *Cntnap2* in mice leads to deficits in core domains of ASD. These mutant mice display increased stereotyped and inflexible behavior such as increased grooming, digging, repetitive circling, impaired reversal learning, and hyperactivity [30,37]. They also exhibit a decrease in social interaction, social preference, and decreased USVs [30,37,51]. Other behavioral phenotypes that are related to ASD comorbidities are cerebellar learning deficits, spatial learning, decreased spontaneous alternation, epileptiform activity, and mild gait phenotype [30,51,52]. *Cntnap2*-knockout mice also show several differences in sensory behaviors of different sensory modalities. These differences include increased acoustic and tactile–acoustic pre-pulse inhibition and increased responses to tactile stimuli, but no difference in acoustic startle response [37,39]. In contrast to the mouse models, *Cntnap2^-/-^* rats display increased acoustic startle responses, in addition, they display increased avoidance to sounds of moderate intensity, and a lack of rapid audio-visual temporal recalibration [53]. Together, these studies indicate that these *Cntnap2^-/-^* rats show differences in sensory processing at both the pre-attentive and perceptual levels. Moreover, knockout mice also exhibit differences in peripheral sensory behaviors underscored by increased mechanical tactile sensitivity to von Frey hair application and pain-related hypersensitivity in response to heat, and chemical algogens [54]. Furthermore, mutant mice perform normally in the buried food test, but they seem to lack a preference for novel odors indicating differences in odor processing or perception [30,55]. Overall, consistent with behavioral differences in autistic people, the *Cntnap2* knockout model displays various ASD-related phenotypes especially seen by deficits in social, repetitive, and restricted behaviors, as well as deficits in several sensory modalities.

The SHANK3 family consists of major scaffolding proteins that play essential roles in the synaptic organization of receptors and cytoskeletal proteins. SHANK3 is essential in the synapse by anchoring to other postsynaptic density proteins including glutamate receptors and neuroligins [56]. The *SHANK3/Shank3* gene has multiple promoters and alternative splicing of the 22 *SHANK3/Shank3* exons leads to multiple *SHANK3/Shank3* isoforms that are highly conserved in mice [40]. This has led to the generation of multiple *Shank3* knockout models with variation in *Shank3* exon deletions. Interestingly, while these *Shank3* models all show ASD-specific behavioral phenotypes, the profoundness of the behavioral alterations seems to relate directly to which exon is deleted [57]. The behavioral phenotypes found in *Shank3* knockout models include stereotypies seen by increased self-grooming and multiple social differences such as deficits in social approach, reciprocal interaction, and abnormal USVs. The *Shank3^-/-^* mice also display behaviors deficits in spatial learning and memory, reduced locomotor activity, impaired motor performance, increased anxiety-like behavior, impaired novel object recognition, and altered striatal-dependent learning [41,42,52,58]. Sensory symptoms found in these mice include altered responses in the tactile domain, such as decreased texture recognition, increased tactile–acoustic pre-pulse inhibition, and increased response to air-puff stimuli [39,59]. Differences in pain perception have also been reported such as impaired heat hyperalgesia in inflammatory and neuropathic pain and increased hotplate sensitivity [34,60]. In addition, these models display altered auditory sensitivity such as decreased acoustic startle response and pre-pulse inhibition [32,41]. Other sensory differences are impaired performance in the buried food test and impaired cerebellar sensory learning [41,58]. Thus, in line with the other behavioral models, a loss of SHANK3 leads to profound behavioral alterations in ASD-related behavior including several differences in responses to sensory stimuli and processing.

### 2.2. Early Behavioral Phenotypes

Although core ASD-related behaviors such as social interaction and stereotyped behaviors have been widely studied in these mouse models, few studies have assessed the developmental trajectory of this behavioral phenotype. The first symptoms in autistic people occur during early life and their developmental trajectories can highly vary [1]. To further understand these developmental trajectories, research is needed with a focus on the early onset of these phenotypes during early development. Monitoring the timing of the behavioral phenotypes could provide valuable information about sensitive windows in phenotype development. Multiple labs assessed social behavior during early life by measuring USVs in *Nrnx1^-/-^* pups from postnatal day (PND) 2–12 [47,48]. They reported reduced ultrasonic vocalization in pups with alterations in the complexity of USVs, indicative of deficits in early communications in pups [2]. Early phenotypes in *Shank3* and *Cntnap2* knockout mice also included a reduced number of USVs [51,58]. These impairments in USVs were also found across several time points during early development from postnatal day PND 3–12. Other social impairments were also found in these models later during development, *Shank3^-/-^* and *Cntnap2^-/-^* mice juveniles display decreased social interactions around PND 21–25 [30,32,42]. Social behavior was also decreased at several moments during early adolescence in *Nrxn1* knockout mice (PND 27–35) and late adolescence in *Shank3* knockout mice (PND 42–56) [47,48,61]. In addition to social behavior, Armstrong et al. also studied other behavioral phenotypes across developmental periods in *Nrxn1^-/-^* and found delays in reaching developmental milestones in juvenile *Nrxn1* knockout mice from PND 5–15 [47]. These delays in developmental milestones include differences in locomotor performance such as geotaxis performance or vertical screen grasp. Such differences can also be found in the *Cntnap2^-/-^* and *Shank3^-/-^* model between PND 4–20 [37,41]. Together, this indicates that several early impairments in developmental milestones and especially in social functioning are present in these genetic models. This suggests that these genes play important roles during neurodevelopment and highlights the importance of studying behavior across developmental trajectories.

**Table 1 genes-13-00028-t001:** Overview of behavioral phenotypes in the *Nrxn1*, *Cntnap2*, and *Shank3* rodent models of synaptic dysfunction in ASD.

Genetic ASD Rodent Model	Sensory Phenotypes	Social Phenotypes	Stereotyped and Repetitive Behaviors	Other Phenotypes	Early Behavioral Phenotypes
*Nrxn1*	↑ Response to startle [31,36]	↑ Aggression [35,47]	↑ Grooming [31]	↑ Locomotion activity [36,38]	PND 2–12 ↓ Number and complexity of USVs [47,48]
	↓ Acoustic prepulse inhibition [31]	↑ Preference for social stimulus versus nonsocial stimulus [35]	↑ Novelty responsiveness and habituation [38]	↑ Motor performance [31]	PND 5–15 ↓ Developmental milestones: body weight, length, vertical screen grasp negative geotaxis, ear canal opening and cliff avoidance [47]
	↓ Eyeblink conditioning [48]	↓ Social interaction [35,47]	↓ Nest building [31,35]	↑ Anxiety [35]	PND 26 ↑ Locomotion [48]
		↓ Preference for social novelty [46]		↓ Instrumental and spatial learning [36]	PND 27–30 ↓ Social interaction during social play [48]
				↓ Reward processing [49]	PND 30 ↑ Object investigation in novel object task [48]
					PND 30 ↓ Social investigative behavior [47]
					PND 30–35 ↓ Performance in food-reward task [48]
					PND 30–35 ↓ Prosocial helping behavior and nurturing behavior [48]
*Cntnap2*	↑ Mechanical sensitivity in the von Frey tests [54]	↓ Three chamber social preference [30,37]	↑ Stereotypic motor movements and behavioral inflexibility [30]	↓ Freezing [37]	PND 3–12 ↓ USVs [51]
	↑ Response in pain sensitivity to algogens [54]	↓ Vocalizations in response to estrous females urine [37]	↑ grooming and digging [30]	↑ Epileptic seizures and epileptiform activity [30]	PND 4 ↑ Rolling on the side during walking [37]
	↑ Acoustic startle responses and moderate-intensity sound avoidance [53]		↑ locomotor activity and gaiting phenotype [30,37]	↓ Morris water maze learning [30]	PND 4–15 ↑ Geotaxis [37]
	↑ Tactile–acoustic prepulse inhibition [39]		↓ Nest building [30]	↓ Spontaneous alternations [30]	PND 21 ↓ Social interaction in juvenile play [30]
	↑ Response to air-puff [39,59]				
	↑↓ Prepulse inhibition [37,53]				
	↓ Audio-visual temporal recalibration [53]				
	↓ Withdrawal latency in the hot plate test [54]				
	↓ Eyeblink conditioning, cerebellar sensory learning [52]				
*Shank3*	↑ Hotplate sensitivity [34]	↓ Preference for social novelty [32,33,34,39,62,63]	↑ Self-grooming [32,33,34,40,41,42,58,61,62,63]	↑ Anxiety-like behavior [32,33,34,39]	PND 4 ↓ Ultrasonic vocalizations [58]
	↑ Tactile prepulse inhibition [39,59]	↓ Preference for social stimulus versus nonsocial stimulus [32,33,39,40,62]	↑ Circling behavior [34,40,41]	↑ Dominance-like behavior [32]	PND 5–13 ↓ Negative geotaxis [41]
	↑ Response to air puff [39,59]	↓ Social interaction [40,58,64]	↑ Repetitive object exploration [40]	↓ Locomotor activity [32,34,41,58]	PND 10–12 ↓ Mid-air righting task [41]
	↓ Eyeblink conditioning, cerebellar sensory learning) [52]	↓ Adult USVs [40,58,64]	↑ Repetitive hole board exploration [40,58]	↓ Motor performance [32,34,40,41,42,58,63]	PND 12–15 ↓ Response auditory startle [41]
	↓ Acoustic startle response [32,41]		↓ Marble burying [34,41]	↓ Object recognition and exploration [34,41,42]	PND 13, 14 ↓ Grasping reflex [41]
	↓ Acoustic prepulse inhibition [32]		↓ Nest building [41]	↓ Barnes maze training and reversal [41]	PND 14, 17–21 ↓ Weight [41]
	↓ Buried food test [41]			↓ Contextual fear testing [41]	PND 15 ↓ Home cage nest preference [58]
	↓ Exploration of nonsocial odors [41]			↓ Spatial learning in the Morris water maze [34,40,62]	PND 15–20 ↓ Wire suspension [41]
	↓ Texture recognition [39,59]			↓ Striatal dependent learning [58]	PND 21–25 ↓ Social interaction [32,42]
	↓ Heat hyperalgesia in inflammatory and neuropathic pain [60]			↓ T-maze reversal [62]	PND 42–56 ↓ Social interaction [61]

## 3. From Neural Circuit to Targeted Gene-Based Therapy

The above-discussed data show a behavioral phenotype of core symptoms of ASD in genetic mouse models of synaptic dysfunction. However, the question remains how and when during development these genes contribute to the observed phenotype. Over the last years, it has become more evident that many genes involved in neurodevelopmental disorders, including ASD, cluster in subnetworks that show commonalities in the spatiotemporal expression trajectories during early neurodevelopment [19]. For example, gene clusters that are involved in synaptic remodeling and functioning (e.g., alternations in postsynaptic density and synaptic transmission) show increased expression during early life. This is a time period when establishing and changing connections are essential for brain development. Moreover, a growing body of literature also suggests that functionally convergent biological processes are modulated through these genes, by altering certain synaptic processes [19]. In the next section, we will discuss where and when during neurodevelopment the *NRXN1*, *CNTNAP2*, and *SHANK3* genes are expressed.

### 3.1. Spatiotemporal Expression of the Synaptic Genes Nrnx1, Cntnap2, and Shank3

The function of the *NRXN1* gene is evolutionary conserved and the gene is widely expressed throughout the brain [24,25,26,65,66]. Specifically, high levels of expression can be found in the cortico-striatal–thalamic circuits [67]. Human cortical *NRXN1* expression increases with age during post conception and is highest during critical periods of brain development [65,66]. Indeed, cortical expression is specifically high during the late embryonic period and early postnatal life (2nd trimester to 3 years of age) [65]. A conditional deletion of *Nrxn1α* in excitatory telencephalic forebrain circuits leads to reward processing dysfunctions in mice, underscoring the importance of *Nrxn1* in those brain regions for certain forms of reward-processing [49]. More recently, Davatolhagh and Fuccillo demonstrated that a loss of *Nrxn1α* leads to divergent synaptic transmission within the dorsal medial striatum for inputs from the dorsal prefrontal cortex and parafascicular thalamic nucleus [67,68]. Taken together, these findings suggest that the loss of *Nrxn1* leads to alterations in cortico-striatal–thalamic circuits where expression of *Nrxn1a* is high during a specific developmental window, namely mid-prenatally and during early childhood.

*CNTNAP2* is expressed in both the brain and spinal cord and is broadly expressed during brain development [69,70]. In mice, gene expression starts prenatally around embryonic day (E) 14 and greatly increases postnatally through adulthood [55,71]. More specifically, expression starts to rise from gestation weeks 18 to 20 [70,72]. This increase can mainly be found in the cortex (frontal and perisylvian regions), as well as in the thalamus, amygdala, striatum, hypothalamus, and regions of the midbrain [30,55,69,70,72]. Therefore, similar to *Nrxn1*, the expression pattern reflects the involvement of the cortico-striatal–thalamic circuitry. In addition, *Cntnap2* is also expressed in brain regions that are involved in sensory processing of every sensory modality and is present in all primary sensory organs [54,55]. Altogether, these observations suggest the importance of *Cntnap2* expression in cortico-striatal–thalamic circuitry and sensory circuits, particularly during the mid-prenatal stages and through infancy and adulthood [54,55,70,71,72].

*SHANK3/Shank3* is expressed in all brain regions, specifically during development [33,73,74]. Expression peaks during critical periods in brain development when synaptogenesis and synaptic maturation takes place [75]. In mice, *Shank3* shows an increase from PND 1 (start of measurement) to approximately PND 28, with a peak expression between 2 and 4 weeks [73,75]. In line with the mouse studies, *SHANK3* showed higher expression post-natally than pre-natally, and this expression remained high during infancy in humans [74]. *Shank3* is highly expressed in the striatum, thalamus, cerebellum, cortex, and hippocampus [33,73]. Brain region-specific disruption of *Shank3* in forebrain circuits leads to excessive self-grooming, while disruption specifically in the striatum leads to repetitive exploratory behaviors in these transgenic mice [76]. However, these specific deletions do not lead to the full phenotype displayed by global *Shank3* deletion including social and learning deficits. This suggests that these brain regions do not cause these social phenotypes by themselves but rather suggest that more complex neural circuits are involved in which both regions are possibly important in these behaviors through cortical–striatal interactions [76]. Moreover, functional defects in striatal synapses and cortico-striatal circuits in the *Shank3* mutant mice have been reported, again highlighting the involvement of those brain areas in *Shank3* pathology [33]. In addition to the brain, *Shank3* is also expressed in the peripheral nervous system [77]. *Shank3* expression was found in primary sensory neurons of the peripheral nervous system in the dorsal root ganglion (DRG) [60]. In mice, a loss of *Shank3* led to reduced transmission in the DRG and these mice demonstrated a loss of sensitivity to capsaicin-induced pain [60]. Therefore, *Shank3* might also play a role in regulating certain sensory modalities through their peripheral function. Overall, *SHANK3* expression peaks during infancy and is highly expressed in cortico-striatal–thalamic brain regions as well as hippocampal, cerebellar, and sensory regions.

Based on the data mentioned above, multiple commonalities in spatiotemporal expression can be found in brain areas that are also highly implicated in the core symptoms of ASD. Recently, the Autism Mouse Connectome (AMC) study compared connectivity alterations in 16 autism-related genetic and etiological models including *Cntnap2* and *Shank3* knockout models, but not *Nrxn1* [78]. They identified a wide variety of connectivity differences in these ASD models, that fits with the heterogeneous character of the clinical presentation and pathophysiological findings observed in ASD. However, based on these connectivity differences the models were grouped into four different clusters, characterized by common atypicalities in specific brain connections. For example, the second cluster was characterized by under-connectivity between cortico-striatal areas and inferior colliculus, but increased connectivity between the ventral orbital areas, lateral septal nuclei cortex, and hippocampus. Interestingly, although the *Nrxn1* mouse model was not included in this study, both *Shank3* and *Cntnap2* gene knockouts were clustering together in the second cluster. This observation suggests a convergence in underlying processes. Moreover, across all clusters, there were also similarities found in affected brain regions. Somato-motor regions, olfactory, and cortical subplate (somatomotor, insular, and striatal) networks show higher vulnerability to abnormal connectivity. This was found to be independent of the direction of the abnormalities (i.e., over- or under-connectivity). These brain regions are linked to the sensory- and motor-related impairments that are frequent in ASD, and commonly altered connectivity in these regions further suggests the critical involvement of these regions in the pathology of ASD. Indeed, the cortico-striatal pathway is implicated to be a mediator of numerous ASD-related behaviors such as stereotyped and repetitive behaviors, rigidness, abnormal reactivity to sensory stimuli, rewards stimuli, and social approach [79]. The observed impaired fronto-striatal–motor connectivity corresponds to brain areas of peak expression during development in the *Nrxn1*, *Cntnap2*, and *Shank3* as mentioned above. Together, these findings suggest that there are commonalities in spatio-temporal gene expression and brain connectivity in cortico-striatal–thalamic brain regions.

### 3.2. Molecular and Downstream Targets of Nrxn1, Cntnap2, and Shank3

The high expression levels during development and, in some cases, throughout adulthood, as well as the connectivity differences in brain areas relevant to the described behavioral differences, implicates a role for altered brain circuit development in these models. This observation raises the question regarding the functions these proteins have in these brain areas. To identify the underlying ASD etiology and identify possible biomarkers, it is important to understand the underlying circuitries that are involved with the behavioral phenotypes. Therefore, in the next paragraphs, we will address the specific functions of the NRXN1, CNTNAP2, and SHANK3 proteins relevant to ASD. 

Neurexins are presynaptic cell-adhesion molecules that are coupled to extracellular postsynaptic cell adhesion molecules and intracellular PDZ domain proteins [44]. This domain can interact with a subunit of the AMPA receptor (GluA1) and with the NMDA receptor (NDMAR), via GKAP and the postsynaptic density (PSD) complex [56]. Furthermore, NRXN1 mutations in humans are also known to impair synaptic transmission [80]. Loss of this gene leads to defects in both excitatory synaptic strength as well as synaptic transmission in hippocampal slices [31]. Furthermore, synaptic transmission is also impaired in the brainstem and neocortex in *Nrxn1* knockout mice [26]. Moreover, they found a decrease in the number of inhibitory synapses that could not account for the much larger impairment of synaptic transmission by itself, suggesting other mechanisms are involved in the impaired transmission. α-Neurexins are essential for Ca^2+^-triggered neurotransmitter release. Indeed, loss of *Nrxn1* leads to impaired neurotransmitter release because of reduced synaptic Ca^2+^ channel function, consequently impairing synaptic transmissions. This suggests an important role for neurexin in the organization of presynaptic machinery [26]. Conditional knockout mice that enable genetic control of alternative splicing of neurexins showed that different neurexins perform distinct functions in transsynaptic control of NMDA and AMPA receptors. Nrxn1 SS4+ alternative splicing selectively enhanced NMDAR-responses, but not AMPAR-responses, while Nrxn3 SS4+ induced suppression in AMPAR-responses, but not NMDAR responses. Therefore, by the trans-synaptic control of NMDA and AMPA receptor responses, the Nrxn1 protein is involved in mediating presynaptic control of postsynaptic responses [81]. Together, neurexin 1 seems to have an important function in excitatory synaptic strength and transmission, through both pre- and postsynaptically influencing neurotransmitter release, and glutamate receptor function. 

The *CNTNAP2*-encoded protein, CASPR2, is a cell adhesion protein that belongs to the neurexin superfamily. It is a presynaptic transmembrane protein with a long extracellular region, to form a trans-synapse binding with contactins, and a short cytoplastic region that can interact with several presynaptic proteins [50]. CNTNAP2 is involved in the myelination of axons and is essential for neural circuit assembly. Indeed, in mice loss of *Cntnap2* leads to disruptions of neural networks caused by decreased dendritic arborization and dendritic spine development in cortical neurons [30,82]. Consequently, this led to an overall decrease in synapse number and reduction in synaptic transmissions. Moreover, Gdalyahu et al., found that the reduced number of dendritic spines was caused by an increase in spine elimination in *Cntnap2^-/-^* mice [83]. Specifically, a reduction in newly formed spines was found through increased elimination of these spines in *Cntnap2* knockout mice. These observations underscore an important role for *Cntnap2* in the stabilization of new synaptic connections. Not surprisingly, the deficits in synaptic connections in *Cntnap2^-/-^* mice led to changes in network activity indicative of a fundamental role for *Cntnap2*^-/-^ in the developing cortex by structural organization of cortical neurons [82]. It should be noted that in addition to changes in excitatory spines, reductions in the number of inhibitory interneurons have also been reported in *Cntnap2^-/-^* mice [30], thus affecting both excitatory and inhibitory synapses and possibly causing changes in E/I balances during development. In excitatory synapses, a loss of *Cntnap2* seems to specifically reduce AMPA receptor trafficking [84]. In addition, hyperactive Akt-mTOR signaling has been found in the hippocampus of *Cntnap2^-/-^* mice [85]. Dysregulation of this pathway has also been implicated in the pathogenesis of other genetic mouse models associated with ASD such as *Tsc1^+/-^*, Tsc2*^+/-^*, *Pten^-/-^* and *Fmr1^-/-^* mice [28,86]. During development, specifically, *Cntnap2* also seems to play a role in neural migration. Indeed, neuronal migration abnormalities can be found in the corpus callosum and layer V–VI of the somatosensory cortex of *Cntnap2* knockout mice indicating that CNTNAP2 is necessary for neuronal migration of cortical projections [30]. Following development, *Cntnap2* plays an important role in myelinated axons of mature neurons. There, it organizes a juxtaparanodal complex by binding to protein 4.1B in the nodes of Ranvier [87,88]. This interaction leads to the recruitment and clustering of voltage-gated potassium channels at juxtaparanodes. Together, these findings indicate that both during development and after development, CNTNAP2 is important in mediating neural migration, synaptic plasticity, and synaptic transmissions. 

SHANK3 is a major scaffolding protein located at the postsynaptic density of excitatory synapses that interacts with key components of synaptic plasticity [56]. The *SHANK3* gene has many alternative promoter and splicing options resulting in various protein isoforms with tissue-specific expression functions. In neurons, SHANK3 links glutamate receptors, NMDARs and AMPARs, to the actin skeleton. This link is established by the interaction of SHANK3 with multiple postsynaptic density proteins and a loss of SHANK3 leads to reduced levels of these PSD proteins including Homer1b/c, PSD95, GKAP, NMDARs, and AMPARs, which together form the so-called glutamate receptosome [33,40,61,89,90]. These are proteins that neurexins also interact with indirectly via PDZ interactions and are important for cytoskeleton regulation, synaptic transmission, and plasticity [44,56]. Specifically, in the striatum and cortex SHANK3 performs an essential function in glutamate receptor signaling by recruiting Homer1b/c to the PSD [62]. The deficits in synaptic transmission and plasticity caused by a loss of *Shank3* include reduced glutamatergic transmission, deficient long-term potentiation (LTP), impairments in mTOR activation, and altered NMDA/AMPA ratios in the prefrontal cortex, cerebellum, and hippocampus [40,61,63,64,90]. Furthermore, *Shank3* deletions also lead to changes in spine morphology and density in striatal and hippocampal neurons. These changes include reduced spine density, increased dendritic length, and increased PSD length and thickness [33,40]. Consequently, a loss of *Shank3* in mice leads to reduced cortico-striatal connectivity and deficits in synaptic plasticity [32,33,56,89] However, although in adult mice cortico-striatal connectivity was reduced, mice at PND 14 exhibit hyperactivity of cortico-striatal circuits [91]. Therefore, SHANK3 seems to be a key player in synaptic transmission by interacting with essential postsynaptic density proteins and a loss of SHANK3 possibly has different effects during early development and adulthood. 

Together, the proteins that these genes encode are essential for synaptic transmission and plasticity and loss of these proteins leads to synaptic deficits and ultimately changes in neuronal networks. In addition, through interacting with PSD proteins, all three genes play an important role in glutamate receptor functioning and transmission. The question remains whether deficits such as changes in glutamate signaling leads to the ASD behavioral phenotypes by acting through the common brain circuits (e.g., cortico-striatal circuitry) at the relevant development window(s). Therefore, local manipulations of the glutamate receptors at the level of cortico-striatal circuits during specific temporal developmental windows will be needed to address this question.

It should be noted that the regulation of synaptic plasticity in part also takes place at the level of regulation of transcription, by processes such as chromatin remodeling, with a central role for the ASD-related genes *MECP2* and *CHD8*. Established *Mecp2* and *Chd8* genetic mouse models exist that display ASD-related behavioral phenotypes such as altered sensory responses as well as showing synaptic and brain connectivity deficits [39,59,78,92,93]. Aligned with the synaptic genes discussed in this review, these genes are specifically important during embryonic and early postnatal development in cortico-striatal neuronal circuits; recently, these topics have been elegantly described in detail for *Mecp2* and *Chd8* by Hoffmann and Spengler et al., 2021; Li and Pozzo-Miller, 2020; Smith et al., 2019, and Varghese et al., 2017 [94,95,96,97].

## 4. From ASD Development to Windows of Treatment Opportunity

Despite the identification of specific proteins, cells, and circuits that are dysregulated in ASD pathophysiology, no validated biomarkers and limited effective symptomatic treatments exist. Most treatments to date show effectiveness for ASD-related comorbidities, but no etiology-directed treatments exist [21]. Adults are the biggest target group for symptom-modifying interventions in ASD. However, because of the developmental nature of ASD, a challenge is the timing of symptom-modifying treatments. A lot of ASD-related genes play important roles during early development when brain circuits are formed. This raises the question of whether altered gene function during those periods specifically leads to altered circuit functioning across domains. Or if early changes in specific circuits lead to behavioral deficits in one domain which causes other circuits to develop differently and consequently leads to the ASD behavior across different domains. This leads to the question of what the effectiveness of treatment is after these important developmental time points. 

### 4.1. Timing of Intervention in ASD

It has been shown that in mouse models for *Shank3,* both early (E18) and adult (2–4.5 months) genetic reinstatement could reverse some of the ASD phenotypes. This includes the rescue of social deficits and repetitive grooming, but not anxiety and locomotion [89,98]. The effectiveness of both early and adult genetic reinstatement indicates the ability and flexibility of reversal of behavioral phenotypes and suggests that late intervention, even post-development, might still be useful to reverse phenotypes. In line with this notion, it should be noted that many ASD-related genes also remain to be expressed throughout adulthood further indicating that these genes might still play a role in post-development processes. However, the effect of *Shank3* reinstatement on the full ASD phenotype was not studied in these *Shank3* mouse models, for example, it is unknown if early or adult reinstatement is also able to rescue the sensory phenotype found in these mice. Therefore, more research on the timing of interventions is needed, as well as on their effects on the full range of behavioral phenotypes. For example, treatment with bumetanide, a drug that alters the synaptic excitation-inhibition (E/I) balance, reduces the severity of ASD symptoms in children and improves ASD-related behavioral outcomes in mice [99,100]. Nonetheless, this treatment has been suggested to be more effective in younger age [100]. Indeed, social responsiveness scores showed a marginal, but significant treatment-by-age effect indicating that younger participants showed more improvement on this outcome [100]. Although more research is needed on the age effect of bumetanide treatment this could relate to the sensitive window of excitation inhibition during development. Additionally, a primary selection by measuring the E/I imbalance in patients could be a basis of stratification for subgroups that possibly benefit from bumetanide or related drugs [101]. Moreover, although one of the bumetanide studies revealed no superior effect of bumetanide over placebo on broad ASD symptoms, it did significantly improve repetitive behaviors [100]. This reveals the possible benefit of stratification by subgroups based on certain behavioral differences including repetitive behavior that would benefit bumetanide treatment [100]. Stratification is essential because of the diversity in ASD. Not all ASD types are caused by monogenetic alterations and only small percentages of autistic people have the same genetic mutations, therefore, single-drug therapy is impossible. Furthermore, because of this genetic heterogeneity, investigation into possible symptom-modifying treatments has focused on common downstream targets of gene clusters for potential personalized treatment approaches. The reviewed literature on *CNTNAP2*, *NRXN1*, and *SHANK3* mentioned above lays out multiple possible molecular targets, such as Akt-mTOR signaling or glutamate receptor signaling.

### 4.2. Treatment of Tactile Sensory System 

The sensory system is a highly conserved system on multiple levels from cell to circuit [102]. While sensory symptoms have been reported in up to 90% of autistic people, around 60% specifically show tactile sensory symptoms [7]. Research on the underlying mechanisms of these symptoms has revealed that in several genetic ASD mouse models, *Cntnap2*, *Shank3*, *Fmr1*, *Gabrb3*, *UBE3A*, and *Mecp2*, display tactile abnormalities [39,59,103]. They show that, in *Mecp2* and *Gabrb3* ASD models, the peripheral mechanosensory neurons, called low-threshold mechanoreceptor neurons and their connections within the spinal cord are dysfunctional due to a loss of GABA_A_ receptor-dependent presynaptic inhibition [59]. Specific treatment of the tactile sensory symptoms by targeting the mechanosensory neurons led to improvement in tactile as well as other core symptoms of ASD such as social improvements [39]. Acute treatment with a GABA agonist that acts directly on the mechanosensory neurons reduces tactile hypersensitivity in six ASD mouse models. While chronic treatment (PND 1–42) in two genetic models, *Shank3* and *Mecp2*, additionally reduced anxiety-like behaviors and improved social behavior [39]. However, not all ASD-related symptoms could be improved, suggesting that mechanosensory neurons contribute to the ASD phenotype, but might not cause a full phenotype by themselves. 

### 4.3. Treatment of Proteins Related to Synaptic Plasticity in ASD

As discussed above mTOR is implicated in humans and multiple models of ASD. The mTOR pathway is involved in the regulation of synaptogenesis, corticogenesis, and associated functions of neurons [28,85,86,104]. In *Cntnap2* mutant mice, the Akt-mTOR pathway is overactivated, specifically in the dorsal root ganglia (DRG), important for the transmission of peripheral sensory information, and hippocampus [85,105]. Inhibition of Akt-mTOR signaling decreases DRG overactivation and rescues mechanical and heat hypersensitivity as well as rescuing social deficits. This implicates the role of the dorsal root ganglia in the behavioral phenotype of *Cntnap2*; however, more research is needed to understand if overactivation in the DRG alone leads to behavioral symptoms [85,105]. In a clinical trial, patients with tuberous sclerosis complex (TSC), a genetic disorder caused by a mutation in the TSC1 or TSC2 gene of which a high percentage (40–50%) of people also have ASD, were treated with the mTOR inhibitor everolimus [106]. This clinical trial showed a positive trend towards improvements of ASD symptoms, while another clinical trial studying the same drug in TSC was not successful [107]. A relatively high and broad age range of the participants in the context of neurodevelopment (i.e., age range 4–17) could possibly have contributed to this difference, and research in only younger individuals might show more robust improvement of the symptoms [108]. Indeed, Amegandjin et al. identified a sensitive window for mTOR inhibitor treatment during the third postnatal week to rescue social behavior in adult *TSc1* mutant mice [109]. This effect is mediated by rescuing parvalbumin interneuron connectivity in these mice [109]. In the cortex, these cells are important in modulating sensory responses and therefore also important for the development of normal social behaviors [39,59]. Abnormalities in these interneurons can also be found in postmortem tissue from people with ASD and in multiple animal models including *Shank3* and *Cntnap2* mutant mice [39,110,111]. More research has to prove if targeted treatment with mTOR inhibitors could be beneficial specifically for a subgroup of people with genetic mutations that are linked to mTOR or parvalbumin interneuron dysfunction during development.

Another potential downstream drug candidate for ASD is the PSD95–GKAP–Shank–Homer complex or ‘glutamate receptosome’. Remodeling of the glutamate receptosome is necessary for the induction of NMDA-dependent plasticity and is required for neuronal plasticity via the activation of mTOR signaling pathways [90]. This glutamate receptosome is interrupted by the *Shank3* mutation in mice, leading to altered plasticity and ASD-like behaviors such as stereotyped behaviors. Early restoration of this complex between PND 6-8 restored the ASD-like deficits *in Shank3* knockout mice [90]. Moreover, pharmacological enhancement of specifically metabotropic glutamate receptor 5 (mGlu5) signaling also reverses the behavioral deficits in *Shank3* knockout mice [62]. Together these studies highlight a novel target for treatment in autistic people that show altered glutamatergic signaling. Altered synaptic transmission, synaptic plasticity, and spine density were also found specifically in striatopallidal medium spiny neurons (MSNs) in *Shank3* mutant mice [112]. This led to deficits in striatopallidal MSNs [112]. By selectively enhancing the striatopallidal MSNs activity, repetitive grooming deficits were reversed, showing that specifically targeting the striatum also has the potential to rescue a behavioral phenotype at least for repetitive behaviors [112]. However, the effect of glutamate receptosome or striatopallidal MSN restoration on social or sensory phenotypes is not yet known. 

As stated in the previous sections, altered synaptic spine density is a common pathological change in people with ASD as well as in genetic mouse models [30,33,40,82,104]. The actin cytoskeleton is essential for the structure and stability of spines and is also crucial for plasticity mediated by NMDA receptor responses [56,61,113]. The SHANK3 protein, in addition to interacting with PSD95–GKAP–Shank–Homer complex, also interacts with actin filaments. In *Shank3* deficient mice, deficits in both social behavior and NMDAR hypofunction are rescued by inhibiting cofilin or activating Rac1, two actin-regulating proteins, in the prefrontal cortex [61]. These results indicate that the aberrant regulation of actin dynamics in the synapse and loss of synaptic NMDARs are involved in the ASD-like phenotype and using these actin dynamics to modify synaptic plasticity could be additional potential targets for treatment [61]. Although the effectiveness of the above-mentioned interventions still needs to be explored in humans, several potential synaptic targets exist that could potentially modify common pathological mechanisms. Additionally, temporal windows for potential optimal effectiveness of interventions need to be taken into account as gene expression of the above discussed synaptic genes peak during early development from prenatal stages through infancy.

### 4.4. Translation of Drug Target Findings from Rodents to Humans 

As mentioned above, rodent models have facilitated new insights into potential drug targets based on underlying mechanisms of ASD. However, the translation of these preclinical findings has sometimes proven to be challenging [114]. This is partly due to human-specific characteristics in neurobiology. Although the neuroanatomy, neural function, and initial neocortical development are highly comparable across mammals, certain aspects are unique to human brain development [115]. Higher-order cognitive functions such as language distinguish humans and rodent models, and therefore suggest additional developmental processes in humans. These additional processes that are unique to human cortical development include cortical expansion (and complexity of the cortical folding), a prolonged developmental period, increased complexity (more diverse types of interneurons), and unique genetics (especially in non-coding RNA) [115]. Together, these differences give rise to more complexity on genomic, cellular, and circuitry level of the human cortex compared to rodents. 

Because of these unique characteristics, the roles of ASD-related genes in human cortical development specifically are less well studied. New developments in preclinical research, such as the use of patient-derived organoids to study human-specific developmental brain processes, could possibly bridge this gap between mouse models and humans by recapitulating underlying mechanisms found in rodent models and confirming the effectiveness of possible treatment targets [115]. Indeed, a *CNTNAP2* cortical organoid model showed cortical overgrowth during embryonic development [114]. In addition, a *16p11.2* ASD cortical organoid model confirmed several alterations in synaptic-related processes, such as changes in actin cytoskeleton and neural morphology, during early human brain development which are in line with findings in *16p11.2* mouse models [116]. Although rodent models provide essential insight into underlying molecular mechanisms in relation to behavioral phenotypes, complementary studies using organoids that have human-specific characteristics can recapitulate findings of rodent models and reveal early human-specific development brain differences in relation to ASD. Since organoids do not allow to study the relationship between ASD gene functions in behavioral and cognitive performance, combining human cellular and rodent system neuroscience strategies could help untangle the complexity in underlying mechanisms of ASD.

## 5. Conclusions

Over the past decades, research advances have increased our understanding of ASD enormously. Despite the progress in molecular and genetic discoveries, the multi-level heterogeneity of ASD, characterized by diversity in genetics, etiology, symptom severity, and developmental trajectories brings a great challenge in understanding the biological underpinnings and predicting treatment outcomes. The identification of ASD subtypes with shared biological etiologies gives valuable new information for personalized intervention strategies that can be studied with the use of genetic rodent models (Figure 1). The temporal–spatial expression pattern of the synaptic genes *NRXN1*, *CNTNAP2*, and *SHANK3* reveal high gene expression levels during early development in cortico-striatal–thalamic regions. These brain areas are relevant for ASD phenotypes and provide targets for brain-region-specific therapies. These synaptic proteins are involved in structural and functional changes in the brain by mediating synaptic plasticity through regulating synapse transmembrane connections, glutamatergic receptor functioning, and through interaction with other postsynaptic density proteins. Correcting these synaptic deficits by manipulation of shared downstream pathways in region-specific neural circuits could therefore restore synaptic functioning and consequently restore neuronal signaling, brain network connectivity and improve behavioral differences. Ultimately a brain region and target-specific intervention approach could improve the quality of life in certain groups of autistic individuals. This will also require the development of novel approaches that allow for targeted and regional-specific interventions in humans.

The high expression levels of these genes during infancy coincide with the early developmental period when the first symptoms arise in autistic people. Because of this strong developmental component in genetics as well as behavioral presentation, intervention outcomes might depend on specific treatment time windows. Therefore, the next step forward is to see if during early developmental periods potential interventions have higher efficacy to modify ASD-related phenotypes. Earlier detection and intervention might not only modify symptoms during early life but also improve the long-term prognosis for autistic people. In this way, the developmental timing of interventions could help optimize treatments outcomes for autistic people. The behavioral studies in genetic mouse models also indicate an early social and developmental phenotype which highlights the importance of studying early behavioral phenotypes in rodents. However, to ensure translatability between humans and animal models, early behavioral phenotypes should be more extensively studied across all behavioral domains of ASD, and by assessing translatable outcome measures longitudinally. More specifically, investigations regarding sensory functions need more focus because of the high prevalence of sensory differences, the early onset, and the predictive value of such symptoms in autistic people. 

In this review, shared mechanisms underlying ASD are described at the level of synaptic genes. However, it should be noted that, by using the same approach to study other clusters of genes, for example, genes involved in the regulation of transcription and translation (*MECP2* and *CHD8*) shared mechanisms could also be identified which can further facilitate the development of new biomarkers for ASD for autistic people with those specific genetic backgrounds. Moreover, in line with the spatiotemporal function of the synaptic genes described in this review, this cluster of translation and transcription genes also converges at the level of the synapse during development, thereby providing opportunities for shared therapeutic approaches. 

Overall, studying genetic rodent models of ASD reveals convergence of synaptic gene functions that are specifically commonly expressed in cortico-striatal–thalamic brain regions during early development. These shared pathophysiological differences could help facilitate the development of potential new targeted gene-based therapy in the near future, which should specifically be considered in the context of developmental time windows.

## Figures and Tables

**Figure 1 genes-13-00028-f001:**
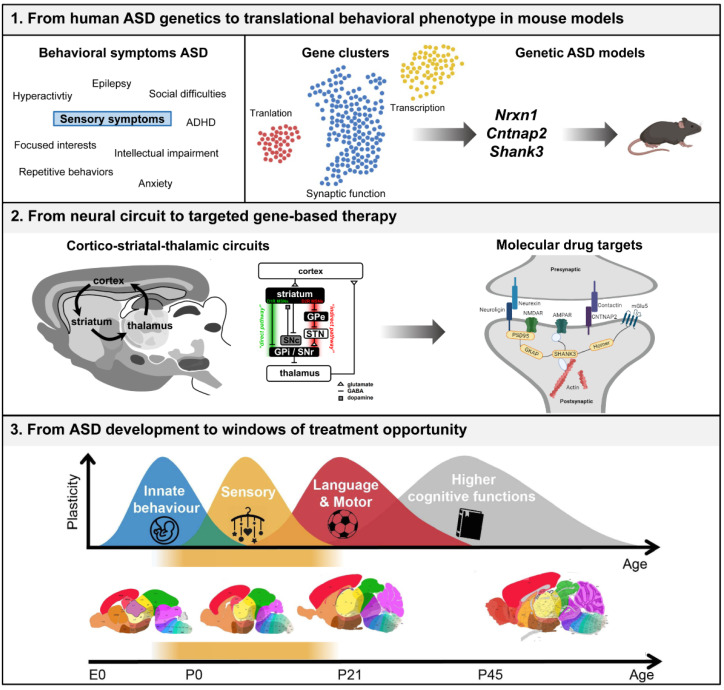
Schematic overview of the development of targeted gene-based therapy for ASD by studying spatiotemporal gene functions in rodent models. Abbreviations: MSN, medium spiny neuron; GPe, globus pallidus pars externalis; GPi, globus pallidus pars internalis; SNc, substantia nigra pars compacta; SNr, substantia nigra pars reticulate; STN, subthalamic nucleus; D1R, D1-type dopamine receptor; D2R, D2-type dopamine receptor. The figure is based on previous reviews, adapted and partly created with BioRender.com [56,117,118,119,120].

## Data Availability

Data sharing not applicable.

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
