# Peer review of "Spatial and Temporal Gene Function Studies in Rodents: Towards Gene-Based Therapies for Autism Spectrum Disorder"

_genes, 2021, doi:10.3390/genes13010028_

Round 1

Reviewer 1 Report

The manuscript „Spatial and Temporal Gene Function Studies in Rodents: towards Gene-based Therapies for Autism Spectrum Disorder“ by Iris W. Riemersma, Robbert Havekes and Martien J. H. Kas, is a review article about the current knowledge on 20 behavioral rodent models of synaptic dysfunction, focusing on behavioral phenotypes, spatial and temporal gene function, and molecular targets that could lead to new targeted gene-based therapy.

Overall, the paper is well-structured, nicely written and gives an in-depth overview describing rodent models of synaptic dysfunction. The manuscript contains 1 figure and 1 table that are explicative of the data presented in the text. I have no major comments regarding this text.

Still the authors’ state that the effectiveness of the mentioned interventions needs to be explored in humans, in my opinion there is a need for a detailed elaboration that human cortical circuitries are substantially different when compared to small laboratory animals as rodents. I would strongly recommend to add a paragraph that will emphasize the complexity and possible uniqueness of human cortical circuitry.

I think that readers interested in rodent models of synaptic dysfunction will benefit by the short paragraph which points on specificities in organization and development of human cortical circuitries.

Author Response

Responses to Reviewer 1

Comment 1: The manuscript „Spatial and Temporal Gene Function Studies in Rodents: towards Gene-based Therapies for Autism Spectrum Disorder“ by Iris W. Riemersma, Robbert Havekes and Martien J. H. Kas, is a review article about the current knowledge on 20 behavioral rodent models of synaptic dysfunction, focusing on behavioral phenotypes, spatial and temporal gene function, and molecular targets that could lead to new targeted gene-based therapy.

Overall, the paper is well-structured, nicely written and gives an in-depth overview describing rodent models of synaptic dysfunction. The manuscript contains 1 figure and 1 table that are explicative of the data presented in the text. I have no major comments regarding this text.

Response to comment 1: We would like to thank the reviewer for the positive response and the constructive comments on our manuscript.

Comment 2: Still the authors’ state that the effectiveness of the mentioned interventions needs to be explored in humans, in my opinion there is a need for a detailed elaboration that human cortical circuitries are substantially different when compared to small laboratory animals as rodents. I would strongly recommend to add a paragraph that will emphasize the complexity and possible uniqueness of human cortical circuitry.

I think that readers interested in rodent models of synaptic dysfunction will benefit by the short paragraph which points on specificities in organization and development of human cortical circuitries.

Response to comment 2: We agree that several aspects of human cortical circuitries and development are substantially different compared to rodents. We elaborated on this in an additional paragraph in the revised manuscript; this paragraph describes similarities between human and rodents, but also addresses the unique features of the human cortex compared to the rodent cortex. Furthermore, this paragraph discusses the implications of these differences for studying ASD genetic rodent models and how to bridge the gap in knowledge that this species difference creates. Therefore, the following paragraph was added to the revised manuscript:

Lines 577-607: 4.4 Translation of drug target findings from rodents to humans                                 “As mentioned above, rodent models have facilitated new insights into potential drug targets based on underlying mechanisms of ASD. However, the translation of these pre-clinical findings has sometimes proven to be challenging [116]. This is partly due to human-specific characteristics in neurobiology. Although the neuroanatomy, neural function, and initial neocortical development are highly comparable across mammals, certain aspects are unique to human brain development [117]. Higher-order cognitive functions such as language distinguish humans and rodent models and therefore suggest addition-al developmental processes in humans. These additional processes that are unique to human cortical development include cortical expansion (and complexity of the cortical folding), a prolonged developmental period, increased complexity (more diverse types of interneurons), and unique genetics (especially in non-coding RNA) [117]. Together these differences give rise to more complexity on genomic, cellular, and circuitry level of the human cortex compared to rodents.                        Because of these unique characteristics, the roles of ASD-related genes in human cortical development specifically are less well studied. New developments in preclinical re-search, such as the use of patient-derived organoids to study human-specific developmental brain processes, could possibly bridge this gap between mouse models and humans by recapitulating underlying mechanisms found in rodent models and confirming the effectiveness of possible treatment targets [117]. Indeed, a CNTNAP2 cortical organoid model showed cortical overgrowth during embryonic development [116]. In addition, a 16p11.2 ASD cortical organoid model confirmed several alterations in synaptic-related processes, such as changes in actin cytoskeleton and neural morphology, during early human brain development which are in line with findings in 16p11.2 mouse models [118]. Although rodent models provide essential insight into underlying molecular mechanisms in relation to behavioral phenotypes, complementary studies using organoids that have human-specific characteristics can recapitulate findings of rodent models and reveal early human-specific development brain differences in relation to ASD. Since organoids do not al-low to study the relationship between ASD gene functions in behavioral and cognitive performance, combining human cellular and rodent system neuroscience strategies could help untangle the complexity in underlying mechanisms of ASD.”

Reviewer 2 Report

The review is well written and clear, but I think that they should mention also the mouse models of Mecp2 and Chd8 mice, given the important role of genetic defects in these two genes in ASD. It would provide a parallel line of analysis for genes involved in chromatin remodelling/regulation of transcription, for which many data are already available also in animal models. This is a major issue of the present review and should be amended.

Author Response

Responses to Reviewer 2

Comment 1: The review is well written and clear, but I think that they should mention also the mouse models of Mecp2 and Chd8 mice, given the important role of genetic defects in these two genes in ASD. It would provide a parallel line of analysis for genes involved in chromatin remodelling/regulation of transcription, for which many data are already available also in animal models. This is a major issue of the present review and should be amended.

Response to comment 1: We would like to thank the reviewer for the positive response, as well as for the constructive comments. We agree that genes that regulate transcription and translation processes such as Mecp2 and Chd8 play very important roles in ASD genetics. Although the focus of this review is directed at genes related to synaptic functions, other genes and gene clusters are also involved in ASD pathology, and as mentioned in the introduction of the review these also include MECP2 and CHD8. Similar to the discussed synaptic genes, such genes also affect synaptic functions amongst having other biological functions. To address the importance of MECP2 and CHD8 in these processes and in ASD, the following text was added to the revised manuscript

Lines 438-447:It should be noted that the regulation of synaptic plasticity in part also takes place at the level of regulation of transcription, by processes such as chromatin remodeling, with a central role for the ASD-related genes MECP2 and CHD8. Established Mecp2 and Chd8 genetic mouse models exist that display ASD-related behavioral phenotypes such as altered sensory responses as well as showing synaptic and brain connectivity deficits [41,60,80,94,95]. Aligned with the synaptic genes discussed in this review, these genes are specifically important during embryonic and early postnatal development in cortico-striatal neuronal circuits; recently, these topics have been elegantly described in detail for Mecp2 and Chd8 by Hoffmann & Spengler et al., 2021; Li & Pozzo-Miller, 2020; Smith et al., 2019 and Varghese et al., 2017  [96–99].”

Lines 645-653: In this review shared mechanisms underlying ASD are described at the level of synaptic genes. However, it should be noted that, by using the same approach to study other clusters of genes, for example, genes involved in the regulation of transcription and translation (MECP2 and CHD8) shared mechanisms could also be identified which can further facilitate the development of new biomarkers for ASD for autistic people with those specific genetic backgrounds. Moreover, in line with the spatiotemporal function of the synaptic genes described in this review, this cluster of translation and transcription genes also converges at the level of the synapse during development, thereby providing opportunities for shared therapeutic approaches.

Round 2

Reviewer 2 Report

The Authors improved significantly the manuscript. I have no other issues to discuss.